# Aldose Reductase as a Key Target in the Prevention and Treatment of Diabetic Retinopathy: A Comprehensive Review

**DOI:** 10.3390/biomedicines12040747

**Published:** 2024-03-27

**Authors:** Alexandra-Ioana Dănilă, Laura Andreea Ghenciu, Emil Robert Stoicescu, Sorin Lucian Bolintineanu, Roxana Iacob, Mihai-Alexandru Săndesc, Alexandra Corina Faur

**Affiliations:** 1Department of Anatomy and Embriology, ‘Victor Babes’ University of Medicine and Pharmacy Timisoara, Eftimie Murgu Square No. 2, 300041 Timisoara, Romania; alexandra.danila@umft.ro (A.-I.D.); s.bolintineanu@umft.ro (S.L.B.); roxana.iacob@umft.ro (R.I.); faur.alexandra@umft.ro (A.C.F.); 2Department of Functional Sciences, ‘Victor Babes’ University of Medicine and Pharmacy Timisoara, Eftimie Murgu Square No. 2, 300041 Timisoara, Romania; 3Doctoral School, ‘Victor Babes’ University of Medicine and Pharmacy Timisoara, Eftimie Murgu Square No. 2, 300041 Timisoara, Romania; stoicescu.emil@umft.ro; 4Field of Applied Engineering Sciences, Specialization Statistical Methods and Techniques in Health and Clinical Research, Faculty of Mechanics, ‘Politehnica’ University Timisoara, Mihai Viteazul Boulevard No. 1, 300222 Timisoara, Romania; 5Department of Radiology and Medical Imaging, ‘Victor Babes’ University of Medicine and Pharmacy Timisoara, Eftimie Murgu Square No. 2, 300041 Timisoara, Romania; 6Research Center for Pharmaco-Toxicological Evaluations, ‘Victor Babes’ University of Medicine and Pharmacy Timisoara, Eftimie Murgu Square No. 2, 300041 Timisoara, Romania; 7Department of Orthopedics and Traumatology, ‘Victor Babes’ University of Medicine and Pharmacy Timisoara, Eftimie Murgu Square No. 2, 300041 Timisoara, Romania; sandesc.mihai@umft.ro

**Keywords:** diabetic retinopathy, aldose reductase, polyol pathway, aldose reductase inhibitor

## Abstract

The escalating global prevalence of diabetes mellitus (DM) over the past two decades has led to a persistent high incidence of diabetic retinopathy (DR), necessitating screening for early symptoms and proper treatment. Effective management of DR aims to decrease vision impairment by controlling modifiable risk factors including hypertension, obesity, and dyslipidemia. Moreover, systemic medications and plant-based therapy show promise in advancing DR treatment. One of the key mechanisms related to DR pathogenesis is the polyol pathway, through which aldose reductase (AR) catalyzes the conversion of glucose to sorbitol within various tissues, including the retina, lens, ciliary body and iris. Elevated glucose levels activate AR, leading to osmotic stress, advanced glycation end-product formation, and oxidative damage. This further implies chronic inflammation, vascular permeability, and angiogenesis. Our comprehensive narrative review describes the therapeutic potential of aldose reductase inhibitors in treating DR, where both synthetic and natural inhibitors have been studied in recent decades. Our synthesis aims to guide future research and clinical interventions in DR management.

## 1. Introduction

Diabetes mellitus (DM) is a multifactorial metabolic disorder characterized by dysregulation of glucose metabolism, originating from inadequate insulin secretion by pancreatic beta cells and/or impaired insulin sensitivity, leading to persistent hyperglycemia and several important complications. The most common complications of diabetes include neuropathy, retinopathy, nephropathy, and cardiovascular disease [1,2]. Approximately 12% of all new cases of severe visual impairment in the United States are caused by diabetic retinopathy (DR), which is the most common cause of blindness in adult patients [1,3]. Moreover, DR is the most common diagnosis for 80% of people who have had DM for more than 20 years. However, it was demonstrated that when effective therapies were given, at least 90% of new patients had their retinal structure and function restored [4]. Since there is currently insufficient effectiveness in screening for DR, the condition can advance unnoticed until irreversible ocular changes have occurred [5]. The pathophysiology of DR is characterized by a cascade of events including vascular endothelial dysfunction, breakdown of the blood–retinal barrier, microvascular occlusion, and ultimately retinal ischemia and neovascularization [6,7,8]. These changes result in a spectrum of clinical manifestations ranging from mild non-proliferative diabetic retinopathy (NPDR) to severe proliferative diabetic retinopathy (PDR), with the latter being associated with the highest risk of vision loss [6]. Several risk factors contribute to the development and progression of DR. Prolonged duration of diabetes, poor glycemic control, hypertension (both systemic and ocular), dyslipidemia, cigarette smoking, and genetic susceptibility are among the most significant factors [7]. Additionally, pregnancy can exacerbate DR, particularly in women with pre-existing diabetes; therefore, pre-conceptional care is needed [9].

Given the dramatic rise in the global prevalence of DM in the past 20 years and the ongoing high incidence of DR in patients with both type I and type II diabetes, screening is essential for prompt detection of patients who are showing symptoms of chronic hyperglycemia-related vision impairment and who need a thorough fundoscopy and proper treatment [10]. Management of DR aims to prevent vision loss by controlling modifiable risk factors, such as hypertension, obesity and dyslipidemia, and treating sight-threatening complications. Tight glycemic control, blood pressure management, and lipid-lowering therapy are essential components of DR management [7]. Additionally, systemic medications—hypoglycemic, hypolipidemic, and antihypertensive drugs—as well as plant-based pharmaceuticals have shown encouraging treatment in the advancement of DR [5,6,10]. Advances in the science of pluripotent stem cells have made it possible to restore retinal functions when these cells are transplanted into animals that have retinal degeneration [11]. In cases of advanced disease, laser photocoagulation, intravitreal injections containing antivascular endothelial growth factor (anti-VEGF) agents, and vitrectomy surgery may be indicated to stabilize or potentially improve vision [12].

Aldose reductase (AR), the first enzyme involved in the polyol pathway, plays an important role in the pathogenesis of DR. Elevated glucose levels in DM lead to increased activity of AR, resulting in the conversion of glucose to sorbitol within retinal cells. Sorbitol accumulation contributes to osmotic stress, formation of advanced glycation end (AGE) products, and depletion of NADPH, exacerbating oxidative damage and cellular dysfunction. Additionally, aldose reductase-derived fructose metabolism leads to the generation of diacylglycerol and activation of protein kinase C (PKC), further promoting low-grade inflammation, vascular permeability, and angiogenesis [13,14,15].

Our purpose was to conduct a comprehensive review focusing on the potential therapeutic efficacy of aldose reductase inhibitors in the treatment of diabetic retinopathy. Through our review, we aimed to critically evaluate the preclinical and clinical evidence supporting the use of synthetic and natural aldose reductase inhibitors as a therapeutic strategy for DR, with the ultimate goal of informing future research directions and clinical practice in the management of DR.

## 2. Pathophysiology

### 2.1. Hyperglycemia, Retinal Microvasculopathy and Metabolic Pathways

Based on its tissue-damaging effects, chronic hyperglycemia is the primary cause of disease development and progression in DR, as reported in the UKPDS [16] and DCCT [17] studies. Individual sensitivity to those effects, however, may be influenced by genetics, and other clinical variables, such as pregnancy, hypertension, and dyslipidemia, have also been linked [3,6,18]. The most important risk factors linked to the onset and progression of DR are described in Table 1.

DR is known worldwide as a microvascular disease. It is believed that hyperglycemia is a key factor in the pathophysiology of retinal microvascular injury, which may lead to irreversible intra- and extracellular changes if untreated [24]. Numerous metabolic processes, such as the polyol path, the formation of AGEs, the PKC pathway, and the hexosamine pathway, have been linked to hyperglycemia-induced vascular injury. These pathways, in conjunction with genetic risk factors, ultimately lead to the activation of growth factors, cytokines and chemokines, which in consequence cause increased vascular permeability, microvascular occlusion, and malfunction of the vascular endothelium [18,25,26,27]. Following microvascular occlusion, retinal ischemia stimulates neovascularization and the development of intraretinal microvascular abnormalities [25].

#### 2.1.1. The Polyol Pathway

Excess glucose is metabolized via overactivity of the polyol route causing irreversible cell damage. Nicotinamide adenine dinucleotide phosphate (NADPH) is used as a cofactor by the retinal enzyme AR to convert glucose to sorbitol. Sorbitol dehydrogenase (SDH), the second enzyme of the pathway, then transforms sorbitol into fructose. [13,14]. Osmotic injury is one of the many negative consequences of sorbitol accumulation in retinal cells [15]. There is insufficient NADPH available for use by glutathione reductase, which is essential for the production of reduced glutathione, as a result of NADPH being used as a cofactor in the first step of the polyol pathway. Subsequently, increased flux in the polyol pathway will cause oxidative stress and ultimately leading to damage of the retinal cells [13].

Early research on the pathophysiology of DR using the polyol pathway was carried out in diabetic animals [28,29]. These investigations demonstrated that the incidence and severity of diabetic retinal lesions in the galactose-fed rats might be decreased by treatment with synthetic aldose reductase inhibitors (ARIs).

Further study has shown that raised AR is localized in many retinal cells, such as glial cells [30], pericytes [31], retinal pigment epithelial cells [32], retinal endothelial cells [32], ganglion cells [30,33], and neurons [33]. These investigations also show that retinal cell degeneration correlates with elevated AR activity. Cell vitality decreased when capillary pericytes or endothelial cells were exposed to higher amounts of glucose or galactose. However, a few articles have emphasized that careful utilization of ARIs prevented cell death and inhibited specific DR changes [34,35].

#### 2.1.2. Advanced Glycation End-Product (AGE) Formation

Although the exact mechanism is still uncertain, increased levels of AGEs in the bloodstream are typically associated with pathologies such as DR, leading to microvascular damage and blood-barrier disfunction. AGEs are heterogeneous compounds formed through non-enzymatic glycation reactions between reducing glucose and macromolecules, such as proteins, lipids, or nucleic acids [36]. In the context of DR, chronic exposure to high glucose levels fosters an accelerated formation and accumulation of AGEs within retinal tissues, making them a key therapeutic target in the potential treatment of DR. AGEs induce structural alterations in retinal proteins through protein cross-linking, thereby perturbing their normal function. Moreover, AGEs trigger a state of increased oxidative stress by generating reactive oxygen species (ROS), which in turn exacerbate pro-inflammatory signaling, microvascular abnormalities and cell death [37]. Furthermore, AGEs modulate the composition of the retinal extracellular matrix (ECM) by directly promoting the synthesis of fibronectin, collagen, and other ECM components. Aberrant ECM remodeling disrupts the structural integrity of retinal blood vessels, fostering fibrosis and exacerbating microvascular dysfunction [38].

Only a small percentage of AGEs that arise in vivo have received thorough characterization and structural definition to date [36]. The rate at which proteins are broken down for glycoxidation, the intensity of hyperglycemia, and the level of oxidative stress in the surroundings are critical elements that determine the development of AGEs [39]. Apart from the endogenous pathway of AGEs, another source is the exogenous path, through smoking and diet [40].

Microvascular alterations, including capillary dropout, microaneurysm formation, and increased vascular permeability, lead to retinal ischemia, vitreous hemorrhage, exudative retinal detachment, and diabetic macular edema. Chronic low-grade inflammation and oxidative stress further exacerbate retinal damage, promoting neurodegeneration, including retinal ganglion cell loss, thinning of the retinal nerve fiber layer, and disruption of the neurovascular unit [25,36].

#### 2.1.3. Protein Kinase C (PKC) Activation

In response to prolonged hyperglycemia, PKC isoforms, particularly PKC-β, are activated within endothelial cells, pericytes, and Müller cells. PKC activation mediates multiple pathological processes, including increased microvascular permeability, endothelial dysfunction, leukostasis (which is believed to be of utmost importance in the beginning of DR), retinal microvascular constriction, and chronic inflammation [24,25]. Furthermore, PKC signaling pathways contribute to the upregulation of VEGF, a key mediator of abnormal angiogenesis and vascular leakage in DR, which can ultimately lead to vitreous hemorrhage and exudative retinal detachment. Consequently, PKC activation exacerbates retinal microvascular abnormalities, promotes prolonged retinal inflammation, and contributes to the progression and complications of DR [24].

#### 2.1.4. The Hexosamine Pathway

In DR, increased flux through the hexosamine pathway occurs due to elevated intracellular glucose levels. This pathway diverts excess glucose metabolites, particularly fructose-6-phosphate, into the production of uridine diphosphate N-acetylglucosamine (UDP-GlcNAc). Elevated UDP-GlcNAc levels lead to hyperglycosylation of proteins, altering their structure and function, and to altered gene expression, ultimately leading to cell apoptosis [24]. This process contributes to the pathogenesis of DR by promoting exaggerated inflammation, oxidative stress, and vascular endothelial dysfunction, exacerbating vascular abnormalities and retinal damage [24,41].

### 2.2. Oxidative Stress

Oxidative stress in DR arises from an imbalance between the production ROS and the antioxidant defense system. Sustained hyperglycemia leads to excess ROS generation within retinal cells, overwhelming antioxidant mechanisms and causing cellular damage. Chronic oxidative stress contributes to damage to macromolecules (DNA, lipids, proteins), vascular dysfunction, inflammation, neurodegeneration, and ultimately vision impairment in diabetic retinopathy. Research on animals has shown that oxidative stress has a role in the development of DR, as well as the metabolic memory phenomenon, which is the resistance of DR changes after adequate glucose levels restored [37].

### 2.3. Inflammation

Inflammation in DR involves the activation of inflammatory pathways due to low-grade chronic hyperglycemia. This triggers the release of pro-inflammatory cytokines-tumor necrosis factor-alpha (TNF-α), interleukin-1β (IL-1β), and interleukin-6 (IL-6) and adhesion molecule intercellular adhesion molecule-1 (ICAM-1) and vascular cell adhesion molecule-1 (VCAM-1), in diabetic animal models and patients, leading to leukostasis, vascular lesions, and recruitment of monocytes and granulocytes into the retina [42]. Inflammatory processes contribute to vascular abnormalities, neurodegeneration, and vision impairment.

### 2.4. Vascular Endothelial Growth Factor (VEGF)

VEGF plays an important role in DR, leading to abnormal angiogenesis and vascular leakage; additionally, it is one of the factors implicated in the development of DME. Hyperglycemia stimulates VEGF production, promoting vascular endothelial cell proliferation, increased vasopermeability and cell migration [40,43,44]. It is currently believed that VEGF functions as an NPDR modulator, and also as a primary PDR initiator. Studies have indicated an association between blood VEGF concentrations and the severity of vascular and retinal changes, as well as the incidence of DR [8]. Nineteen patients with NPDR and twenty patients with PDR had their VEGF concentrations tested in the study of Jain et al. The results were compared to those of nineteen diabetics without ocular disease and nineteen healthy controls. With VEGF, the degree of retinopathy significantly increased [45].

### 2.5. Retinal Neurodegeneration

Retinal neurodegeneration is one of the first phenomena in the development of DR and involves progressive damage to ganglion cells, amacrine cells, and photoreceptors, due to chronic hyperglycemia and associated metabolic disturbances. This event manifests as thinning of the retinal nerve fiber layer (RNFL), loss of neuronal function, and cell apoptosis [44,46]. Contributing factors include oxidative stress, inflammation, altered neurotrophic support, and impaired neurotransmission. Retinal neurodegeneration in DR contributes to vision loss and other visual impairments independent of vascular changes, highlighting that this process may be an independent risk factor [46].

## 3. Biology and Characterization of Aldose Reductase

Aldose reductase, a cytosolic enzyme, is present in various tissues throughout the body, including the retina and presents important substrate specificity [40]. It is particularly abundant in tissues exposed to high glucose concentrations, such as the kidneys, lens, and myelin sheath tissues, and a key factor in developing several of the most important DM complications. In the retina, aldose reductase is found in various cell types, including retinal endothelial cells, retinal capillary wall pericytes, retinal glial cells (Müller cells), and neuronal cells. The enzyme is encoded by the *AKR1B1* gene, and its expression and activity can be mediated by variable factors such as glucose levels, oxidative stress, and several inflammatory cytokines (such as IL-8, IL-6, TNF-alpha) [44,45].

Aldose reductase is a monomeric protein composed of a single polypeptide chain, which folds into a globular structure with multiple alpha-helices and beta-sheets. Aldose reductase also contains a substrate-binding site, where aldose substrates, such as glucose, bind for enzymatic conversion to their corresponding sugar alcohols (sorbitol) [47]. Additionally, the enzyme may have active allosteric-binding sites that modulate its activity in response to regulatory molecules [48]. The first stage of the two-step polyol cycle of glucose metabolism is the reduction of glucose to sorbitol (catalyzed by AR and requires NADPH), which is then transformed into fructose. Several tissues, such as the kidneys, Schwann cells, and retina, lack the enzyme SDH, which raises concerns about the sorbitol pathway. Because of this, sorbitol builds up to dangerous amounts in diabetics or people with exceptionally high glucose levels. Both SDH and AR activities must be strictly controlled at optimal level [49]. With chronic high glucose levels, the antioxidant capacity of cells is diminished by increased AR activity. These processes can increase the production of ROS, which will induce oxidative stress and are thought to be the primary initiating factor of diabetes complications [50] (Figure 1).

## 4. Measurement of Aldose Reductase

Measurement of aldose reductase in DR involves various techniques aimed at assessing its activity and expression levels within retinal tissues or biological solutions obtained from patients. Enzymatic assays represent a direct method for quantifying AR activity, typically involving the detection of the conversion of glucose to sorbitol, by the increased activity of aldose reductase in DM patients [9,51]. Immunohistochemical staining techniques [52,53] and spectrophotometric tests may allow quantification of aldose reductase expression [54]. Gerhardinger et al. have studied the mechanism of action and effects of sorbinil and aspirin using immunohistochemistry and real-time PCR (RT-PCR) [55]. A large study has also used spectrophotometry in detecting AR inhibition of medicinal plants in the potential treatment of DR [56].

Many studies have used cultures of human RPE cells for the detection of AR activity [57,58]. Results have shown that AR activity was highest when RPE cells were cultured in pathophysiological levels of glucose. Additionally, Western blot test allows for the quantification of aldose reductase protein levels in human ocular tissues treated with lysis buffer, providing information about its expression levels [52]. There have been a few studies using Western blot for the determination of AR in various tissues before and after treatment with an ARI [53]. Quantitative real-time PCR (qRT-PCR) is a accurate and efficient molecular investigation [59], which can be utilized to measure aldose reductase mRNA levels in the retina, as well as in the cornea, ciliary body, iris and lens [60]. RNA sequencing analysis can also demonstrate the downregulation of cytokine and inflammatory-related genes following AR inhibition [54,61]. An uniquely developed immunoassay technique that uses a particular antibody directed against AR can measure the enzyme’s concentration [58]. In a study it was shown to strongly correspond with the activity of AR that was extracted from the same people’s erythrocytes [62].

These measurement techniques contribute to the understanding of the role of AR in diabetic retinopathy pathogenesis. Elevated AR activity and expression have been implicated in the development and progression of retinal complications associated with DM, leading to visual impairment [13,31]. By measuring AR levels, researchers can gain insights into the molecular mechanisms underlying DR, including the polyol pathway-mediated osmotic stress, oxidative damage, and cellular dysfunction. Furthermore, assessing AR levels may aid in identifying potential therapeutic targets for intervention in DR [52,57].

## 5. Treatment Targeting Aldose Reductase

Inhibition of AR represents a therapeutic strategy for DR aimed at inhibiting the consequences of elevated polyol pathway activity. Pharmacological inhibition or inactivation of the AR gene lowers AR activity and protects the retina from retinal barrier leakage and neovascularization [63,64]. Furthermore, hyperglycemia increases the flux of glucose metabolism via the polyol pathway, which raises the risk of inflammation and oxidative stress. Retinal inflammation and glucose-induced oxidative stress are avoided when AR inhibitors block the polyol pathway [63,65]. Several potential approaches have been explored to inhibit aldose reductase activity in the context of DR (Figure 2).

### 5.1. Aldose Reductase Inhibitors

Small-molecule inhibitors targeting AR have been developed and investigated as potential therapeutic agents for DR. These ARIs competitively inhibit the enzymatic activity of aldose reductase, thereby reducing the conversion of glucose to sorbitol and attenuating downstream polyol pathway-mediated complications. Clinical studies on ARIs show that a therapeutic response can be induced by partially inhibiting ALR2 [66].

A variety of structural classes of ARIs have been developed: (1) derivatives of carboxylic acids (e.g., Epalrestat and Alrestatin); (2) derivatives of spirohydantoins and other cyclic amides (e.g., Sorbinil, Minalrestat, and Fidarestat); and (3) phenolic derivatives (e.g., related to Benzopyran-4-one and Chalcone). Epalrestat is the sole inhibitor among these that is now marketed for sale. Furthermore, several additional ARIs, including Ranirestat and Sorbinil, had progressed to the late stages of clinical testing and were determined to be safe for use in humans [67,68]. Reducing the variables that contribute to DR’s progression is the most important step in managing the condition. Effective AR inhibition requires potent ARIs because AR inhibition may regulate the factors involved [65]. Historically, there were two types of ARIs that were widely used: spirosuccinimide and carboxylic acids. Subsequently, it was discovered that oral active pyridazinones exhibited more AR selectivity than other drugs [69]. There has been some evidence that administering ARIs to an animal model of diabetes at the beginning of the disease can help avoid diabetic retinopathy [35,70,71,72].

Treatment with ARIs may also inhibit the basement membrane from thickening, as demonstrated in diabetic rat models studies [73]. A study by Hattori et al. showed that leukocyte adherence to endothelial cells mediated by adhesion molecules may be inhibited by adding an ARI to a diabetic rat model [35]. Administration of ARIs has been demonstrated to inhibit two hallmark processes of diabetic retinopathy: an increase in vascular permeability and a breakdown of the blood–retinal barrier [68,74,75,76]. According to recent research, sorbitol and fructose inhibition are both necessary to provide positive outcomes in diabetic retinopathy [68].

Based on long-term studies, Epalrestat was an effective medication in preventing the development of diabetic retinopathy in its early stages [77]. Cunha-Vaz et al. also showed that, when compared to a placebo group, treatment with ARIs for six months significantly decreased the rapid penetration of fluorescein in individuals with early stages of diabetic retinopathy [78].

The studies of Kakehashi et al. have demonstrated inhibition of the polyol pathway in the course of DR in SDT rats by Fidarestat, an ARI [79]. Fidarestat inhibited the amount of VEGF in the ocular fluid and stopped significant fluorescein leakage around the optic disc. They also studied the effect of Ranirestat in the development of fundus changes in DR and observed the inhibitory effect on VEGF levels [80]. Other ocular and non-ocular DM complications, such as cataract and diabetic neuropathy were also studied, confirming that patients under treatment with Ranirestat had a more positive outcome [81]. Other study considered the possibility that ranirestat decreases the retinal thickness in SDT rats. Considering all present and past results, they highlighted that ranirestat might inhibit vascular permeability and minimize the posibility of DME [82]. Other research regarding Fidarestat has observed that the treatment reduced the amount of pericytes, basement membrane thickness, and number of microaneurysms; at a higher dose, it even supressed fundus changes. Fidarestat also prevented the build-up of sorbitol in the retina in a dose-dependent manner [70].

Over a relatively long period, Epalrestat was demonstrated to supress the onset or to diminish the progression of diabetic neuropathy, retinopathy, and nephropathy. However, it was uncertain if Epalrestat’s inhibition of AR, the maintenance of diabetic neuropathy, or a combination of the two is responsible for the prevention of the advancement of diabetic retinopathy/nephropathy [83]. Diabetic neuropathy may act as a potential trigger for the onset or advancement of diabetic retinopathy, diabetic nephropathy, and other microvascular complications, demonstrating intrincacies between these three vascular complicationd of DM. An ARI prevented upregulation of genes in the transforming growth factor (TGF)-β pathway and apoptosis in the retinal vasculature of diabetic rats, as demonstrated by Gerhardinger et al. [55].

A highly selective ARI of a new structural class, pyridazinones, demonstrated efficacy in preventing or reversing early retinal abnormalities in experimental diabetic retinopathy. ARI-809 showed promising results in inhibiting retinal polyol accumulation, improving survival, inhibiting cataract development, normalizing retinal sorbitol and fructose levels, and protecting the retina from abnormalities associated with DM [68]. Another relatively new topical ARI inhibitor, Kinostat, has been studied with the purpose of inhibiting diabetic cataract formation in diabetic animals. Although effective in animals, further studies are still needed for human use [84].

Another study has identified small-molecule ARIs that may be employed in the synthesis of diabetic treatment medications. By choosing specific inhibitors, they suggested that compared to ALR1 and AKR1B10, these may be more potent against the ALR2 protein [81].

### 5.2. Phytocompounds

Several phytocompounds have been investigated for their potential therapeutic effects in DR. These compounds, derived from different plants, possess various bioactive properties that may help inhibit retinal damage and prevent vision loss. Research has demonstrated that phytochemicals’ antiangiogenic, antioxidant, and anti-inflammatory properties may inhibit the advancement of DR [50,85]. Numerous plants have been shown to have AR inhibiting effect and to be employed in the management of diabetic complications, representing a safer option that synthetic compounds [56,70]. A phytocompound that possesses both antioxidative and AR inhibitory activity may be more efficient than one that only possesses only one of these characteristics [86]. Numerous AR inhibitors and antioxidants are derived from plants, including *G. lucidum*, *A. indica*, *T. cordifolia*, *O. sanctum*, *B. ciliata* and *R. arboreum* [56]. When *Ocimum Sanctum* is used in combination with vitamin E, it protects against DR [87].

Julius et al. explored the potential of using natural phytocompounds such as quercetin, kaempferol, and ellagic acid as aldose reductase inhibitors for the treatment of DR. The research suggests that these novel ARIs derived from natural sources could offer a promising and alternative approach to managing DR, showing even higher potency than Epalrestat [66]. Quercetin has been reported to attenuate retinal inflammation and vascular leakage in diabetic animals, potentially through inhibition of the NF-κB signaling pathway [88].

When hesperidin, a flavonoid derived from citrus fruits, was delivered to STZ-induced rats, superoxide dismutase (SOD) expression was elevated and aldose reductase activity and malondialdehyde were decreased [89]. Genistein, a naturally occuring compound belonging to the isoflavones, could inhibit the increased level of aldose reductase caused by hyperglycemic conditions in ARPE-19 cells, from the retinal pigment epithelial cell line [90]. Quercitin and Genistein have also been found to delay the development of diabetic cataract, another major ocular complication [91,92]. Isoflavones derived from Caesalpinia pulcherrima have been demonstrated to lower oxidative stress and inhibit AR activity in DR [93].

### 5.3. Combination Therapy

Given the multifactorial nature of DR, combination therapies targeting multiple pathways involved in its pathogenesis, including AR inhibition, have been investigated. Combining ARI with other pharmacological agents targeting oxidative stress, inflammation, vascular dysfunction, or neuroprotection may offer synergistic effects and improved therapeutic outcomes in DR. Combining ARIs with neuroprotective agents, such as nerve growth factor (NGF) or brain-derived neurotrophic factor (BDNF), addresses both aldose reductase-mediated neurodegeneration and other previously mentioned neurotoxic pathways in the early stages of DR [94].

Anti-inflammatory drugs, such as corticosteroids or non-steroidal anti-inflammatory drugs (NSAIDs), targets both aldose reductase-mediated inflammation and other inflammatory pathways implicated in DR. Aspirin [95], Meloxicam (COX-2 inhibitor) [96], Nepafenac (COX-1 and COX-2 inhibitor) and Etanercept (receptor of TNF-α) [97] have all been studied and have been known to protect against DR microangiopathy and damage to retinal vascularization. Combination therapy might reduce retinal inflammation and suppress vascular dysfunction and neurodegeneration. Antivascular endothelial growth factor (VEGF) agents, such as ranibizumab or aflibercept, addresses both aldose reductase-mediated vascular abnormalities and VEGF-induced neovascularization and vascular leakage in DR; therefore, dual administration of ARI and anti-VEGF substances might inhibit retinal complications, preventing PDR, macular edema and DR evolution [98].

A clinical investigation recently evaluated the role of the angiotensin II antagonist candesartan in both type 1 and type 2 DM with or without early DR, based on promising preclinical studies [99]. Overall, this medication showed promise in slowing the course of DR and, in some cases, reversing it as determined by the ETDRS scale [100,101].

The chemokine ligand CCL2.279 is a powerful mediator of inflammation and the deterioration of the blood retinal barrier. A CCR2/5 receptor antagonist was studied on DR patients [102]. In patients with persistent diabetic macular edema (DME), a clinical trial using infliximab, a monoclonal antibody antagonist of TNF-α, showed a significant improvement in VA and a general reduction in retinal thickness [103]. Since TNF-α stimulates the expression of AR [104], combined therapy with antagonists of TNF-α and ARIs, might be of important value in the treatment of DR.

In addition to the current end-stage strategies for treating DR, cell-based therapy may offer an exciting new approach. In order to improve vascular repair, reverse ischemia, lessen hypoxic/inflammatory stimuli, and stop the advancement of these diseases to their late, sight-threatening phases, this strategy is intended to target the early/intermediate stages of vasodegeneration [105]. In diabetic CD34+ cells, transient suppression of TGF-β1 may be a promising therapeutic approach to restore vascular reparative activity and may improve the chances of effective cellular therapy [106]. As ARIs may prevent upregulation of genes in the TGF-β pathway and apoptosis of retinal cells [55], more detailed studies regarding a combination therapy might be of use.

### 5.4. Gene Therapy

Gene therapy approaches aimed at modulating aldose reductase expression or activity within the (*AKR1B1*) genetic variant was revealed to be the most strongly related with DR in a recent meta-analysis of over 160 candidate gene studies reported for DR [96]. However, many of the studies have not taken into account risk factors in the development of DR. By delivering gene constructs encoding inhibitory RNA molecules or regulatory elements targeting aldose reductase expression, it may be possible to achieve sustained inhibition of aldose reductase activity and ameliorate retinal damage associated with DM. A few promising articles showed that gene therapy combined with AR inhibitors may sustain retinal ganglionar cell survival and even axon regeneration; however, there is no reported data on visual restoration [53]. Polymorphisms in the AR gene have been connected with susceptibility to DR as well as other pathologies linked to DM [107,108]. These connections have been studied in DM type I and II and in several ethnic groups [108]. The recognition of the susceptibility genes may aid in the development of future treatment.

Preclinical studies utilizing gene therapy to inhibit aldose reductase expression or activity have shown promising results in animal models of diabetic retinopathy. These studies have demonstrated improvements in retinal function, preservation of retinal structure, and attenuation of vascular abnormalities following gene-based aldose reductase inhibition [59]. While these preclinical findings are encouraging, translating gene therapy approaches targeting aldose reductase inhibition into clinical practice for DR requires further investigation. Challenges such as optimizing vector delivery, ensuring target specificity, and assessing long-term safety and efficacy in human subjects need to be addressed. RNA sequencing analysis demonstrated that, following AR inhibition, cytokine and inflammatory associated genes are downregulated, which is in line with studies showing TNF-α downregulation and linking AR inhibition to TNF-α driven signaling. Through anti-inflammatory reactions, AR inhibition temporarily protects the retina and optic nerve by postponing retinal ganglionar cells death and axon degeneration. Axon regrowth cannot be aided by AR inhibition alone. Combining an AR inhibitor with the well-known axon regeneration promoters shKLF9/Sox11 may have a synergistic effect on axon regeneration [56].

## 6. Benefits and Limitations

Many studies highlight the long-term etiology of DR and the significantly lower success rate of pharmaceutical interventions that inhibit fundus changes once they are already present, compared to preventative treatment before early and potentially irreversible retinal alterations occur [109]. Therefore, if the clinical research design does not take into consideration known risk factors for DR, efficacy for a certain treatment may not be able to be established, even if the therapeutic premise for AR inhibition is valid. A few studies have investigated treatment with ARI as a tool in the prevention, reversal or delay in the onset and progression of DM complications [110]. Since achieving generalized glycemic control is extremely challenging, more research into developing effective, selective, and non-toxic ARIs is warranted. Safe medications are necessary for long-term DM complications. Since these medications would prevent rather than cure diabetes, they have to be given to all individuals with the condition. Since vitamin C has been shown to normalize the level of sorbitol in red blood cells, regardless of changes in diabetes management, it has been suggested that an ARI may be associated with ascorbic acid for the prevetion and treatment of diabetic complications. Several three-component associations (ascorbic acid, ARIs, and unsaturated fatty acids) have been patented for the treatment of DM complications due to a synergistic mechanism [111]. There is also a belief that a diet with ARIs from natural sources might have the same potency in the prevention of DM complications [111]. While this is convenient and has also shown a promising effect against AR, further studies are needed to evaluate the quantities of dietary ARIs at which we could expect these effects.

Overall, while ARIs offer a targeted approach for reducing polyol pathway-mediated damage in diabetic retinopathy with many potential benefits (Table 2) [13,15,40,69,73,112], their use is associated with certain disadvantages, including limited efficacy, systemic side effects, cataract formation, hypoglycemia risk, and the lack of long-term data [113,114]. Despite the promising results seen in laboratory settings, clinical trials have had mixed success in demonstrating long-term efficacy and safety of ARIs for the treatment of diabetic retinopathy [82]. Despite these challenges, novel ARIs with higher selectivity and fewer side effects, such as ARI-809 [68], have shown promising results in preclinical studies. Continued research and development of new ARIs with enhanced selectivity and safety profiles remain vital to advancing the field of diabetic retinopathy treatment [115]. Moreover, several studies have emphasized the effects of natural compounds against AR, paving a new way into DR treatment.

## 7. Conclusions

AR’s pathogenic role in diabetes mellitus has been established through many intensive investigations. Several ARIs have been used in the treatment of diabetes mellitus and its complications. However, since the majority of studied ARIs were withdrawn due to unfavorable side effects, primarily from the ARIs’ non-specific binding to the aldo–keto reductase family of proteins, which shares a high degree of structural similarity with AR, the identification of novel ARIs has become increasingly important in research. Moreover, research has shown that many natural compounds possess AR inhibitory effects, paving the way to a guided treatment using both lifestyle/diet changes and pharmacologic therapies.

## Figures and Tables

**Figure 1 biomedicines-12-00747-f001:**
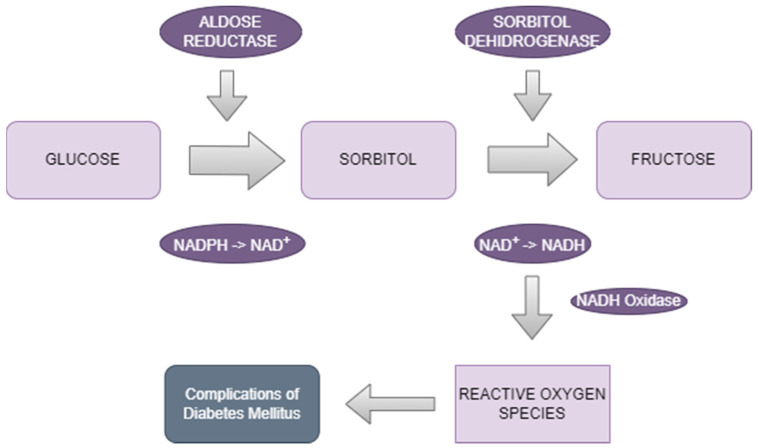
The polyol pathway.

**Figure 2 biomedicines-12-00747-f002:**
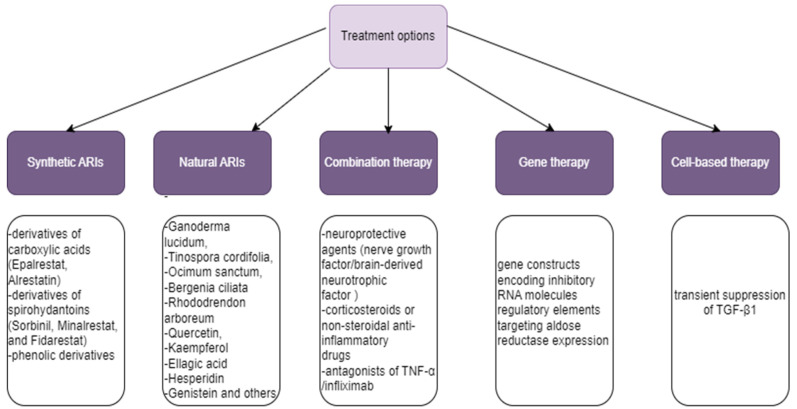
Treatment options targeting aldose reductase inhibition.

**Table 1 biomedicines-12-00747-t001:** Risk factors for the onset and progression of diabetic retinopathy.

Risk Factor	Description
Duration of diabetes [17]	well-known risk factor for the development and progression of DR
Glycemic control [16,17]	poor glycemic control (high HbA1C) is linked to increased risk of DR
Intraocular/systemic blood pressure [16,19]	systemic hypertension might lead to progression of DR through increased vascular permeability and microvascular damage
Dyslipidemia [20]	elevated levels of triglycerides, low-density lipoprotein cholesterol (LDL-C), and reduced levels of high-density lipoprotein cholesterol (HDL-C) managed through diet modification and lipid-lowering drugs
Obesity [21]	elevated body mass index (BMI) and abdominal adiposity leads to the release of cytokines, adipokines and to insulin-resistance (which is thought to be an independent risk factor) weight management is essential
Cigarette smoking [22]	a modifiable risk factor can compromise retinal microvascular integrity and function
Genetic predisposition [23]	an important number of genetic variants have been studied can be managed by targeted therapy in high-risk patients

**Table 2 biomedicines-12-00747-t002:** Potential benefits of using aldose reductase inhibitors (ARIs) for treating diabetic retinopathy.

Decreased production of sorbitol and fructose, which reduces oxidative stress and preserves the antioxidant defense system within the retina
Protection against neuronal apoptosis, glial reactivity, and complement deposition, all of which contribute to retinal damage
Reduction in microaneurysms, basement membrane thickness, and vascular permeability
Blockage of the expression of various pro-inflammatory cytokines and growth factors
Potential prevention of initial focal laser therapy, especially in cases of DME
Prevention of cataract development
Promising outcomes with novel ARIs, which demonstrate higher selectivity and fewer side effects compared to earlier generations of ARIs

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
