# Peer review of "Aldose Reductase as a Key Target in the Prevention and Treatment of Diabetic Retinopathy: A Comprehensive Review"

_biomedicines, 2024, doi:10.3390/biomedicines12040747_

Round 1
Reviewer 1 Report
Comments and Suggestions for Authors
The article “Aldose Reductase as a Key Target in the Prevention and 2 Treatment of Diabetic Retinopathy: A Comprehensive Review” by Danila et al., reviewed the therapeutic potential of AR inhibitors (synthetic and naturally occurring) in mitigating diabetic retinopathy. Authors in this review have provided elaborate information and mentioned very briefly about the therapeutic modalities available in targeting AR. The review is written well and organized meticulously.
Comments and suggestions
It would have been more interesting if the authors have covered the therapeutic sections much more extensively rather than just over-viewing the topic, in particular, the gene therapy and immunotherapies. Since there are many previous articles (reviews) published on the similar aspect, it would have been more interesting if the authors have covered aspects related to modern methods of treatment.
Evid Based Complement Alternat Med. 2022; 2022: 9624118.
Published online 2022 Mar 21. doi: 10.1155/2022/9624118
PMCID: PMC8959960
PMID: 35356240
Inhibition of Aldose Reductase by Novel Phytocompounds: A Heuristic Approach to Treating Diabetic Retinopathy
Angeline Julius, 1 Remya Rajan Renuka, 1 Waheeta Hopper, 2 P. Babu Raghu, 3Sharmila Rajendran, 4 Senthilkumari Srinivasan, 4 Kuppamuthu Dharmalingam, 5Amer M. Alanazi, 6 Selvaraj Arokiyaraj, 7 and S. Prasath 8
Front. Mol. Biosci., 20 November 2023
Sec. Biological Modeling and Simulation
Volume 10 - 2023 | https://doi.org/10.3389/fmolb.2023.1271569
Exploring potent aldose reductase inhibitors for anti-diabetic (anti-hyperglycemic) therapy: integrating structure-based drug design, and MMGBSA approaches
Muhammad Shahab1 Guojun Zheng1* Fahad M. Alshabrmi2
Mohammed Bourhia3*
Gezahign Fentahun Wondmie4 Ahmad Mohammad Salamatullah5
Authors have not covered anything about the cell-based therapies (such as CAR-T cells based therapies). For example, Cell-Based Therapies for Diabetic Retinopathy by Lynn C. Shaw, Matthew B. Neu, and Maria B. Grant, Curr Diab Rep. 2011 Aug; 11(4): 265–274.
In addition, the authors would have provided more analysis on the preclinical and clinical studies in this area of research.
Furthermore, it would have been more appealing if the authors have represented the existing therapies as an overview figure.
The recommendation
Therefore, I recommend this article for publication only after addressing my above mentioned comments
Comments on the Quality of English Language
English is fine. No major concerns
Author Response
Dear reviewer,
Thank you for taking the time to review our manuscript. Your feedback has helped improve our manuscript and we sincerely appreciate your work. I will explain the changes point by point:
- We have explained and even added a few more paragraphs in the treatment section. Since in the manuscript, we have only discussed about aldose reductase inhibition, we could not add information regarding unrelated treatment options, but we have added all the references you suggested and a few more, describing combination therapies that might help inhibition of aldose reductase activity
- We have also added a figure summing up all relevant information regarding treatment options
- Regarding treatment, we have also added a paragraph about prevention of diabetic retinopathy using current treatment therapies
Thank you once again for your time and attention
Reviewer 2 Report
Comments and Suggestions for Authors
Danila et al. presented a comprehensive review in diabetic retinopathy (DR) that highlights the potential of aldose reductase inhibitors, both synthetic and natural, in DR treatment, aiming to inform future research and clinical practices. The manuscript is well written, however, there are some minor aspects the authors should consider revising.
1. In section 3, where aldose reductase was introduced, the authors should include the fact that retina lacks significant activity of sorbitol dehydrogenase, which makes it more susceptible to damage induced by sorbitol accumulation in DM.
2. In section 6, the authors raise a valuable point about when AR inhibition should be administrated to achieve clinical efficacy. This merits further expansion, at least in the existent section, or in a separate section regarding future direction.
Author Response
Dear reviewer,
Thank you for taking the time to review our manuscript. Your feedback has helped improve our manuscript and we sincerely appreciate your work. I will explain the changes point by point:
- In section 3, where aldose reductase was introduced, the authors should include the fact that retina lacks significant activity of sorbitol dehydrogenase, which makes it more susceptible to damage induced by sorbitol accumulation in DM. Response: we have added information regarding the lack of SDH in the retina, thank you so much
- In section 6, the authors raise a valuable point about when AR inhibition should be administrated to achieve clinical efficacy. This merits further expansion, at least in the existent section, or in a separate section regarding future direction. Response: we have detailed this in the section of limitations. We mentioned all relevant information we could find, there are not many studies regarding prevention with the help of know treatments; however, we highlighted that combination therapy has proven to have efficacy in small studies.
- We have also added a figure with treatment options as well as a few other therapy options
Once again, we thank you for your time and attention
Round 2
Reviewer 1 Report
Comments and Suggestions for Authors
Authors have addressed all the comments to the satisfaction